# Beyond COVID-19 Diagnosis: Prognosis with Hierarchical Graph Representation Learning

## Abstract

Coronavirus disease 2019 (COVID-19), the pandemic that is spreading fast globally, has caused over 34 million confirmed cases. Apart from the reverse transcription polymerase chain reaction (RT-PCR), the chest computed tomography (CT) is viewed as a standard and effective tool for disease diagnosis and progression monitoring. We propose a diagnosis and prognosis model based on graph convolutional networks (GCNs). The chest CT scan of a patient, typically involving hundreds of sectional images in sequential order, is formulated as a densely connected weighted graph. A novel distance aware pooling is proposed to abstract the node information hierarchically, which is robust and efficient for such densely connected graphs. Our method, combining GCNs and distance aware pooling, can integrate the information from all slices in the chest CT scans for optimal decision making, which leads to the state-of-the-art accuracy in the COVID-19 diagnosis and prognosis. With less than 1% number of total parameters in the baseline 3D ResNet model, our method achieves 94.8% accuracy for diagnosis. It has a 2.4% improvement compared with the baseline model on the same dataset. In addition, we can localize the most informative slices with disease lesions for COVID-19 within a large sequence of chest CT images. The proposed model can produce visual explanations for the diagnosis and prognosis, making the decision more transparent and explainable, while RT-PCR only leads to the test result with no prognosis information. The prognosis analysis can help hospitals or clinical centers designate medical resources more efficiently and better support clinicians to determine the proper clinical treatment.

## 1 Introduction

Coronavirus disease 2019 (COVID-19) has resulted in an ongoing pandemic in the world. To control the sources of infection and cut off the channels of transmission, rapid testing and detection are of vital importance. The reverse transcription polymerase chain reaction (RT-PCR) is a widely-used screening technology and viewed as the standard method for suspected cases. However, this method highly relies upon the required lab facilities and the diagnostic kits. In addition, the sensitivity of RT-PCR is not high enough for early diagnosis (Ai et al., 2020; Fang et al., 2020). To mitigate the limitations of RT-PCR, the computed tomography (CT) has been widely used as an effective complementary method, which can provide medical images of the lung area to reveal the details of the disease and its prognosis (Huang et al., 2020; Chung et al., 2020), for which RT-PCR cannot. Additionally, CT has also been proved to be useful in monitoring the COVID-19 disease progression and the therapeutic efficacy evaluation (Rodriguez-Morales et al., 2020; Liechti et al., 2020).

The chest CT slices of a patient have a sequential and hierarchical data structure. The relationship between slices possesses more information than the order of the slices. The adjacent ones with the same abnormality could be considered as one lesion. The slices containing the same type of lesions may not be continuous as the lesions are distributed in various lung parts. We propose a diagnosis and prognosis system that combines graph convolutional networks (GCNs) and a distance aware pooling, which integrates the information from all slices in the chest CT scans for optimal decision making. Our major contributions are three-fold: (1) Owing to the sequential structure of CT images, this is the first work to utilize GCNs to extract node information hierarchically, and conduct both diagnosis and prognosis for COVID-19. The prognosis can help facilitate medical resources, e.g., ventilators or admission to Intensive Care Units (ICUs), more efficiently by triaging

mild or severe patients. (2) A novel pooling method called distance aware pooling, is proposed to aggregate the graph, i.e., the patient's CT scan, effectively. The new pooling method integrated with GCNs can aggregate a densely connected graph efficiently. (3) The new model can localize the most informative slices within a chest CT scan, which significantly reduces the amount of work for radiologists.

## 2 RELATED WORK

**AI-assisted and CT-based COVID-19 Diagnosis and Prognosis.** Although RT-PCR is the standard way for COVID-19 diagnosis, there are many limitations using RT-PCR along, e.g., time delay in receiving an RT-PCR test, occurrence of false negatives, and no prognostic information provided, etc. CT images are often recommended as an alternative for precise lesion detection (Alizadehsani et al., 2020). However, as each CT scan includes a large number of (up to several hundreds of) image slices, it requires much time and labor of the radiologists (Shoeibi et al., 2020). Furthermore, since the radiological appearances of COVID-19 are similar to other types of pneumonia, radiologists need to go through extensive training before they can achieve high diagnostic accuracy (Shi et al., 2020).

Recently, several AI-assisted and CT-based COVID-19 diagnostic systems have been developed. Chen et al. (2020) use Unet++ (Zhou et al., 2018) to segment infectious areas in the lung. Butt et al. (2020) develop a deep learning model to detect lesions from CT images, and then use 3D ResNet to classify the images into COVID-19, influenza-A viral pneumonia or healthy groups. Song et al. (2020) use the whole lung for diagnosis instead of only extracting the lesions. Wang et al. (2020) propose an AI system that can diagnose COVID-19 patients as well as conducting the prognostic analysis.

**Graph Neural Networks.** Various types of graph neural networks (GNNs) have been proposed, which can be divided into spectral or non-spectral domains. In the spectral domain, the Fourier transformation and graph Laplacian define the convolutional filters (Bruna et al., 2013). By utilizing Chebyshev polynomials, Kipf & Welling (2016) simplify the filters of graph convolution, rendering a layer-wise propagation method. However, the generalization of spectral methods may not be ideal due to the variety of graphs (Bronstein et al., 2017). Non-spectral methods focus on the local topology of nodes, directly working on graphs instead of the Fourier domain. Methods proposed by Hamilton et al. (2017), Monti et al. (2017) and Veličković et al. (2017) aggregate nodes based on adjacent nodes when the next layer is created. This aggregation process, as mentioned by Gilmer et al. (2017), can be regarded as a message-passing process.

**Pooling Methods** Pooling methods allow GNNs to hierarchically aggregate nodes, obtaining and assembling local information of graphs. The major purpose of the hierarchical pooling method is to use a locally based model to aggregate nodes in each layer, so that a higher level graph representation can be created (Lee et al., 2019). The self-attention based pooling method (Mao et al., 2018) is implemented for video classification, locally obtaining weighted and fused feature sequences. Spectral pooling methods, such as the one proposed by Ma et al. (2019), focus on the application of eigen-decomposition to capture the graph information. However, since spectral pooling methods are computationally demanding, they may not be accessible to large graphs. Non-spectral pooling methods are scalable to large graphs and pay more attention to the local structures of graphs. Adaptive Structure Aware Pooling (ASAP) (Ranjan et al., 2020) and DiffPool (Ying et al., 2018) resort to clustering techniques, aggregating nodes into different clusters, and then choosing top clusters based on cluster ranking scores (Gao & Ji, 2019). However, the ASAP method partially ignores the edge weight information when using hops to aggregate nodes, resulting in unstable convergence.

## 3 METHODOLOGY

We propose a GCN-based diagnosis and prognosis method that models the sequential slices of CT scans hierarchically. To downsample and learn graph-level representation from the input node features, a novel distance aware pooling method is proposed. In this paper, the node features refer to the slices in a CT scan. The model gradually extracts information from the slice level to the patient level by graph convolution and pooling. Eventually, a higher-level representation is learned, and further used for diagnosis, prognosis, and lesion localization. The schema of our model is illustrated in

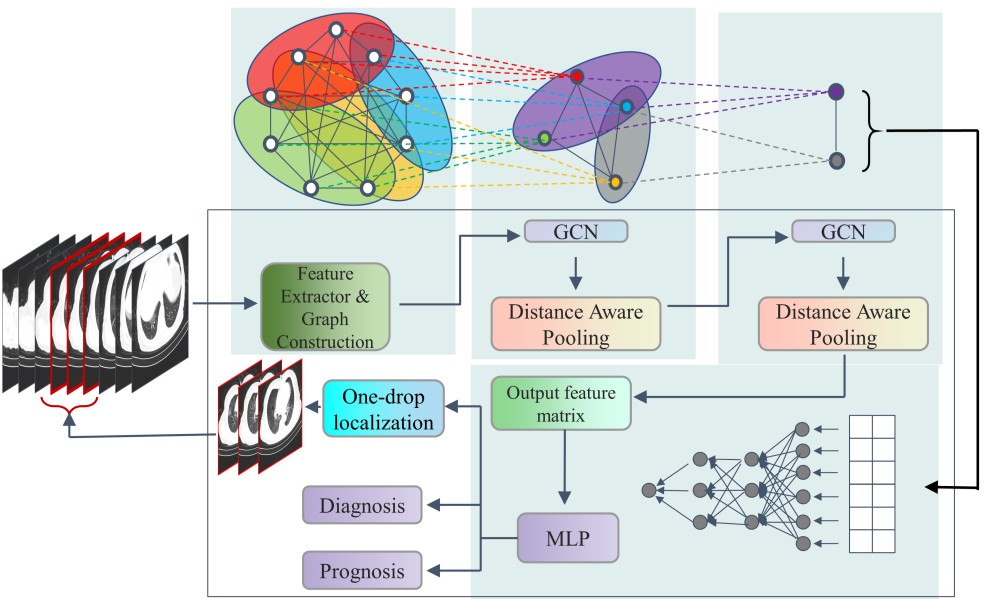

Figure 1: Schema of our model structure. The CT scan of one patient is converted to a densely connected graph. The GCN and the Distance Aware Pooling method are integrated to learn a graph-level representation. At each of the two layers, the node embedding are learned by GCN, and the cluster membership is calculated by the Distance Aware Pooling method. The aggregated graph, which is on the top right corner, is passed to a MLP. Meanwhile, the one-drop localization can localize the most informative slices in the CT scan in a weakly supervised manner.

Figure 1, which is composed of GCNs, pooling modules, a multilayer perceptron (MLP) classifier, and a one-drop localization module. The graph convolution-based method can integrate all slices in the chest CT scans for optimal decision making. Furthermore, we propose the one-drop localization to localize the most informative slices, so that radiologists may focus on those recommended slices with the most suspected lesion areas. Consequently, the proposed model can produce visual explanations for the diagnosis and prognosis, making the decision more transparent and explainable. We argue that this method could effectively assist radiologists by reducing redundancies in the vast amount of CT slices during diagnosis and prognosis in the clinical settings.

### 3.1 PROBLEM STATEMENT

Let $\mathcal{G}\,(\mathbb{V}, \mathbb{E})$ be a patient's CT scan graph, with $|\mathbb{V}| = N$ nodes and $|\mathbb{E}|$ edges, where $|\cdot|$ represents the cardinality of a set. For each $v_i \in \mathbb{V}$, $\boldsymbol{x}_i$ is the corresponding $d$-dimensional vector. Let $\boldsymbol{X} \in \mathbb{R}^{N \times d}$ be the node feature matrix, and $\boldsymbol{A}^{adj} \in \mathbb{R}^{N \times N}$ be the weighted adjacency matrix. Each entry in $\boldsymbol{A}^{adj}$ is defined based on cosine similarity, which is $\boldsymbol{A}^{adj}_{i,j} = <\boldsymbol{x}_i, \boldsymbol{x}_j>/(\|\boldsymbol{x}_i\| \cdot \|\boldsymbol{x}_j\|)$. Define the distance matrix $\boldsymbol{A}^{dis}$ as $\boldsymbol{A}^{dis} = \mathbb{1} - \boldsymbol{A}^{adj}$, where $\mathbb{1}$ is the matrix with all entries of 1's. The definition of $\boldsymbol{A}^{dis}$ means that the more similar two nodes are, the shorter distance they have. In the construction of $\boldsymbol{A}^{adj}$, the diagonal entries are automatically 1's since $i = j$ is allowed.

Each graph $\mathcal{G}$ has a label $y$. For diagnosis, the label represents its class from normal, common pneumonia, and COVID-19. For prognosis, the class indicates whether a COVID-19 positive patient develops into sever/critical illness status. Thus, the diagnosis and prognosis of COVID-19 is a task of graph classification under our setting. Given a training dataset $\mathbb{T} = \{(\mathcal{G}_1, y_1), \ldots, (\mathcal{G}_M, y_M)\}$, the goal is to learn a mapping $f : \mathcal{G} \to \boldsymbol{y}$, which classifies a graph $\mathcal{G}$ into the corresponding class $\boldsymbol{y}$. Our model is composed of two modules: $f_1$ includes node convolution and feature pooling, and $f_2$ involves a MLP classifier which determines the class of each graph. At each layer of $f_1$, node embeddings and cluster membership are learnt iteratively. The first module can be written as $f_1 : \mathcal{G} \to \mathcal{G}^p$, where $\mathcal{G}^p$ is the pooled graph with fewer nodes and a hierarchical feature representation. The second module is $f_2 : \mathcal{G}^p \to \boldsymbol{y}$, which utilizes the graph-level representation learnt for patient

diagnosis and prognosis. The two modules are integrated in an end-to-end fashion. That is to say, the cluster assignment is learnt merely based on the graph classification objective.

## 3.2 NODE CONVOLUTION AND FEATURE POOLING

### 3.2.1 NODE CONVOLUTION

Node convolution applies the graph convolutional network to obtain a high-level node feature representation in the feature matrix $X$. Although several methods exist to construct the convolutional network, the method recommended by Kipf & Welling (2016) is effective for our case, which is given by $X^{(l+1)} = \sigma(\sqrt{D^{(l)}} A^{adj(l)} \sqrt{D^{(l)}} X^{(l)} W^{(l)})$, where $D^{(l)}$ is the diagonal degree matrix of $A^{adj(l)} - I$, and $W^{(l)} \in \mathbb{R}^{d \times k}$ is learnable weight matrix at the $l$-th layer. Due to the application of feature pooling, the topology of the graph changes at each layer, and thus the dimension of matrices involved are reduced accordingly.

### 3.2.2 DISTANCE AWARE POOLING METHOD

We propose an innovative pooling method, which includes graph-based clustering and feature pooling. Below, we outline the pooling module and illustrate how it is integrated into an end-to-end GCN based model. Empirically, it is shown to be more robust for densely connected graphs. The overall structure of the pooling method is illustrated in figure 2.

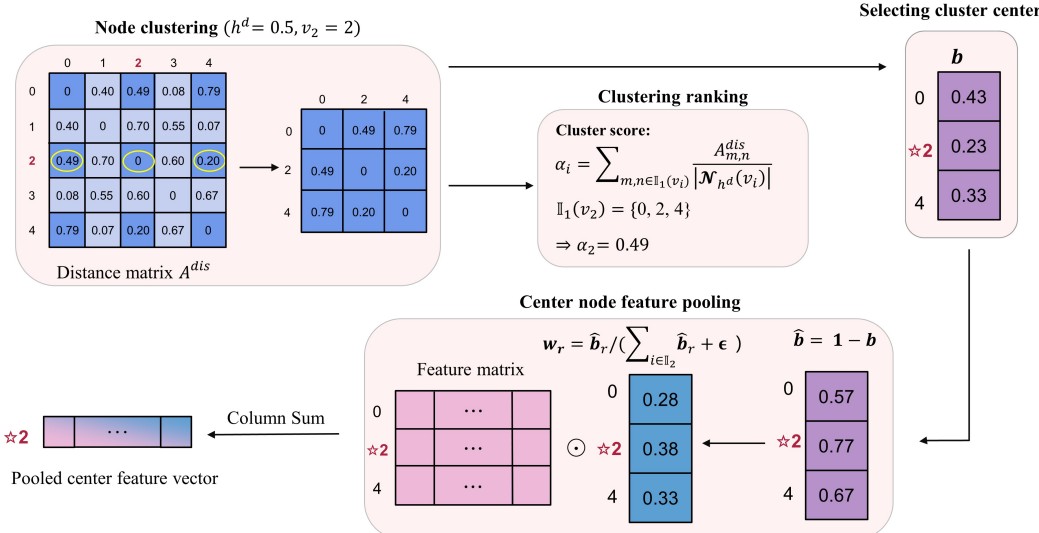

Figure 2: The structure of the Distance Aware Pooling method with a numeric example. Four steps mentioned in the above figures are node clustering, clustering ranking, selecting cluster centers, and center node feature pooling, which correspond to those mentioned in Section 3.2.2. In this figure, the proposed center in the node clustering step is node $v_2 = 2$, and the corresponding $RF^d$ neighborhood within $h^d = 0.5$ are nodes 0, 2, and 4. It should be noted that the proposed center may not be the final center for this cluster. The cluster score $\alpha_i$ of each cluster with proposed center $v_i = i$ can be calculated. With $\alpha_i$, the top $k$ proportion clusters could be decided. In the selecting cluster centers step, $b$ is the average of each row sum, and node $v_2$ is still chosen as the center with the least average distance value. In the center node feature pooling step, $w_r$ is the weight of nodes derived from $b$. In this way, we can derive the weighted pooled center feature vector.

**Improved receptive field.** The concept of the receptive field, $RF$, used in the convolutional neural network (CNN) was extended to GNNs (Ranjan et al., 2020). They defined $RF^{node}$ as the number of hops required to cover the neighborhood of a given node, such that given a chosen node, a cluster can be obtained based on a fixed receptive field $h$. However, this design may not be applicable to densely connected graphs, because one node may be connected to most of the nodes in the graph. Even given a small value of $h$, for instance, $h = 1$ which is the default value (Ranjan et al., 2020), clusters formed within hop $h = 1$ may include most of the nodes of the graph. Hence, the clustering

step may be inefficient. In addition, the value of hops can only be integers, which loses the edge weight information in the clustering process.

Therefore, we define an improved receptive field for densely connected graphs $RF^d$ denoted by $h^d$, which can be treated as a radius centered at a given node. The value of $h^d$ is not restricted to integers but can be a positive real number. Define $\mathcal{N}(v_i)$ as the local neighborhood of the node $v_i$, and $\mathcal{N}_{h^d}(v_i)$ as the $RF^d$ neighborhood of the node $v_i$ with a radius $h^d$, and $\forall v_j \in \mathcal{N}(v_i)$, $v_i$ and $v_j$ are connected by the edge $(v_i, v_j)$.

**Node clustering.** Inspired by the clustering and ranking ideas mentioned in Ranjan et al. (2020) and Gao & Ji (2019), we propose a local node clustering, and score ranking method. Each node is considered as a center of a cluster for a given $h^d$. Then, we score all the clusters and choose the top $k$ proportion of them to represent the next layer's nodes with pooled feature values, where $k$ is a hyperparameter.

**Clustering ranking.** Given a node $v_i$ and a radius $h^d$, $\mathcal{N}_{h^d}(v_i)$ is the corresponding $RF^d$ neighborhood. Let $\mathbb{I}_1(v_i)$ be the index set of the nodes in $\mathcal{N}_{h^d}(v_i)$. The cluster score is defined as $\alpha_i = \sum_{m,n \in \mathbb{I}_1(v_i)} \boldsymbol{A}^{dis}_{m,n}/|\mathcal{N}_{h^d}(v_i)|$, where $m \neq n$. If $\alpha_i$ is small, nodes in $\mathcal{N}_{h^d}(v_i)$ are close to each other. The top $k$ proportion of clusters form the next layer's nodes.

**Selecting cluster centers.** We first introduce some notation: $\forall v_j \in \mathcal{N}_{h^d}(v_i)$, define $\mathbb{V}_j = \mathcal{N}[v_j] \cap \mathcal{N}_{h^d}(v_i)$ as the set of nodes connected to $v_j$ in $\mathcal{N}_{h^d}(v_i)$, where $\mathcal{N}[v_j]$ is the closed neighborhood of the node $v_j$. Let $\mathbb{I}_2(v_j)$ be the index set of the nodes in $\mathbb{V}_j$. The node score is defined as $b_j = \sum_{c \in \mathbb{I}_2(v_i)} \boldsymbol{A}^{dis}_{j,c}/|\mathbb{V}_j|$. The node with the smallest value of $b_j$ is chosen as the center of the cluster, and is used to represent a new node in the next layer's node representation.

**Center node feature pooling.** Based on the node scores, we rank the nodes in $\mathbb{V}_j$ and assign a weight $w_r$ to $\boldsymbol{x}_r$, where $r \in \mathbb{I}_2(v_j)$. Let $\hat{\boldsymbol{b}} = \boldsymbol{1} - \boldsymbol{b}$. The weight vector $\boldsymbol{w}$ is defined as $w_r = \hat{b}_r/(\sum_{i \in \mathbb{I}_2(v_j)} \hat{b}_i + \epsilon)$, where $\boldsymbol{1}$ is a vector with all 1's, $\epsilon$ is an extremely small positive value to avoid 0's in the denominator, and $\boldsymbol{b}$ is the vector containing all the weights of nodes in $\mathbb{V}_j$. The value of the pooled center feature is defined as $\boldsymbol{x}^p = \boldsymbol{X}_r \boldsymbol{w}$, and then $\boldsymbol{x}^p$ is used as the center feature representing $\mathcal{N}_{h^d}(v_i)$.

**Next layer node connectivity.** Following the idea in Ying et al. (2018), the connectivity of the nodes in the next layer is preserved as follows. According to the above ranking and pooling methods, a pooled graph $\mathcal{G}^p$ with the node set $\mathbb{V}^p$ is obtained. The next step is to decide the pooled adjacency matrix $\boldsymbol{A}^{adj,p}$. Define matrix $\boldsymbol{S}$ such that the columns of $\boldsymbol{S}$ are the top $k$ clusters' weight vectors $\boldsymbol{w}$. Hence, the pooled adjacency matrix is defined as $\boldsymbol{A}^{adj,p} = \boldsymbol{S}^T \boldsymbol{A}^{adj} \boldsymbol{S}$.

## 3.3 CLASSIFICATION AND LOCALIZATION BASED ON POOLED GRAPHS.

**Graph classification.** A hierarchical representation for each patient is abstracted by the above GCNs with distance aware pooling. The representation is an $N' \times D'$ feature matrix, where $N'$ is the number of clusters and $D'$ is the number of features in each cluster. For each feature, the mean and the maximum over the $N'$ clusters are calculated. Subsequently, we obtain a $2D'$ feature vector. The first $D'$ elements are the mean values of each feature, and the second $D'$ elements are the maximal values. Then, we use an MLP to classify the $2D'$ feature vector into one of the three classes: COVID-19, common pneumonia, or normal. The MLP classifier is mainly composed of two linear layers. For each input $2D'$ feature vector, it can output the scores of each class, normalized by the softmax function.

**Weakly supervised informative CT slices localization.** Besides the classification, localizing the CT slices with lesions is also vital for the diagnosis. To achieve this goal, we propose the one-drop localization to select the most informative CT slices for the model to make a decision. This method does not require mask annotation for lesion, and thus the localization is learned in a weakly supervised manner. We are inspired by the stepwise regression that is used for knowledge discovery (Cios et al., 1998), an automatic procedure that integrates the variable selection. One of the ideas is backward elimination. For each patient with the $N' \times D'$ feature matrix, we first predict the target class using the above MLP classifier. Then, we drop one cluster each time so that we obtain $N'$ new feature matrices in total with an equal size of $(N'-1) \times D'$. For each of the new feature matrices,

the score of the target class is calculated using the same MLP classifier. The lowest score from the $N'$ results and the corresponding dropped cluster are chosen. Since ignoring this cluster leads to the lowest score of the target class, this cluster should contain the most crucial diagnosis information. We can then determine the center of this cluster, and trace back to the corresponding input CT slice. We further localize the top $k^s$ CT slices with the highest similarity to the cluster center. We believe this whole $k^s + 1$ CT slices are most informative for the diagnosis, which can be suggested as a reference to the radiologists.

## 4 EXPERIMENTS AND RESULTS

For diagnosis and prognosis, we compare our model with the baseline, a 3D ResNet-18 classification network (Zhang et al., 2020), and the state-of-the-art graph classification method, GCNs with ASAP (Ranjan et al., 2020). Moreover, we appraise whether the proposed method can deliver meaningful and interpretable clusters on the input chest CT scans by comparing the localization results with the slices containing lesions.

### 4.1 DATASET

We utilize the CT dataset from the 2019 Novel Coronavirus Resource (2019nCoVR) (Zhang et al., 2020). The dataset includes the complete chest CT scans of 929 COVID-19 positive patients, 964 common pneumonia patients, and 849 healthy individuals.

The dataset also provides these patients' clinical prognosis, whether the patients developed into severe/critical illness status, referring to the admission to ICU, mechanical ventilation, or death. The prognosis analysis could support the hospitals to designate medical resources more efficiently. In addition, the dataset summarizes the slices with lesions for COVID-19 positive and common pneumonia patients, which can be used to evaluate the one-drop localization method. Each CT slice is normalized into the dimension $256 \times 256$. For each chest CT scan, we use systematic sampling to ensure that an equal number of slices are chosen.

### 4.2 DATA PREPROCESSING: CHEST CT SCAN IMAGE FEATURE EXTRACTION

**CNN feature extraction.** Feature extraction is a common strategy in computer vision. In our case, we utilize Inception V3 (Szegedy et al., 2016) pretrained on ImageNet (Russakovsky et al., 2015). The feature map of the bottleneck layer, which is the last layer before the flatten operation, is regarded as the node representation in the graph. In our case, each node represents a slice in a chest CT scan, and the node representation is a vector of dimension $2048$.

**Wavelet decomposition extraction.** Considering each slice in CT scans as a 2-dimensional signal, it can be viewed as a function with two variables, which can be reconstructed as a summation of wavelet functions multiplying their coefficients for a given resolution (Mallat, 1989). The formula required to reconstruct the image is

$$f(x,y) = \frac{1}{\sqrt{MN}} \sum_{m,n} \left[ W_\varphi(j_0, m, n)\varphi_{j_0,m,n}(x,y) + \sum_{i=H,V,D} \sum_{j=j_0}^{\infty} W_\Psi^i(j,m,n)\Psi_{j,m,n}^i(x,y) \right],$$

where $f(x,y)$ is the energy function of an image, $M$ and $N$ are the image dimension, $W_\varphi$ and $W_\Psi$ are the coefficients of scaling function $\varphi$ and wavelet function $\Psi$ respectively. The coefficient $W_\varphi$ is used in down streaming analysis since it contains the major approximation information of an image. We choose the Haar wavelet function with resolution 3. The flattened approximation matrix of the image signal, which is a vector of dimension $1024$, is used as the feature embedding of a slice in a CT scan.

### 4.3 IMPLEMENTATION DETAILS

For dataset 19nCoVR, we use systematic sampling to ensure that 48 number of slices for each CT scan are chosen. CT scans of 60% individual are randomly chosen as the training set, 25% as the test set, and the remaining 15% for the validation. To avoid information leakage, the dataset is split

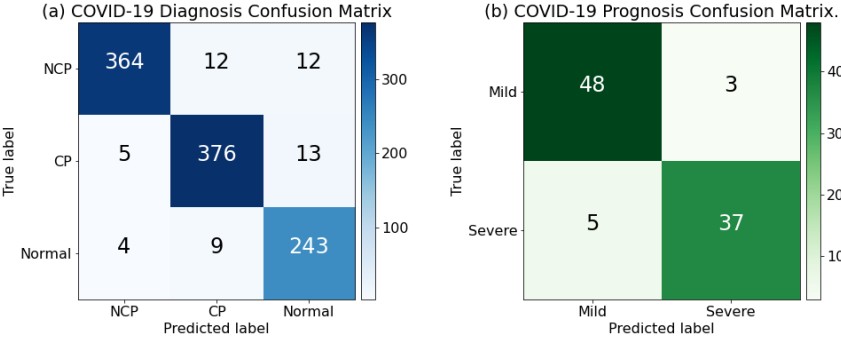

Figure 3: The confusion matrices for COVID-19 diagnosis and prognosis. 'NCP', 'CP', and 'Normal' indicate COVID-19 positive patients, common pneumonia patients, and healthy individuals respectively. 'Severe' or 'Mild' indicates whether a patient develops into severe/critical illness status. The severe/critical illness status refers to admission to the ICU, mechanical ventilation, or death.

according to individuals instead of the CT scan. Similar to Zhang et al. (2019) and Ying et al. (2018), we repeat the data splitting aforementioned on 20 random seeds. For each random seed, the model is trained from scratch. The maximum, average and standard deviation of test accuracies are reported.

We use five GCN layers, and the MLP classifier is composed of two fully connected layers. The pooling proportion $k$ is set as 0.5 for the experiments of diagnosis and prognosis. The Adam optimizer with an initial learning rate 0.0002, and a linear decay schedule is used. The negative log-likelihood loss is used for training the diagnosis and prognosis model. For prognosis, the parameters of GCN and pooling are initialized using those pretrained on diagnosis task. All models are trained for 128 epochs with early stopping applied.

## 4.4 QUANTITATIVE RESULTS

(a) **Performance Evaluation of COVID-19 Diagnosis**

| Method | Feature Extractor | Average Accuracy ± SD | Best Accuracy | Time (s/epoch) |
|---|---|---|---|---|
| GCN-DAP | Inception V3 | **93.93% ± 0.41%** | **94.80%** | 22.70 |
| GCN-DAP | Wavelet | 83.65% ± 1.01% | 85.16% | 20.90 |
| GCN-ASAP | Inception V3 | 75.20% ± 18.70% | 93.74% | 30.00 |
| GCN-ASAP | Wavelet | 51.43% ± 11.90% | 81.50% | 27.25 |
| GCN-DiffPool | Inception V3 | 71.22% ± 23.73% | 94.31% | 18.35 |
| GCN-HGP-SL | Inception V3 | 93.89 % ± 0.39% | 94.22% | 45.60 |

(b) **Performance Evaluation of COVID-19 Prognosis**

| Method | Feature Extractor | Average Accuracy ± SD | Best Accuracy | Time (s/epoch) |
|---|---|---|---|---|
| GCN-DAP | Inception V3 | **82.70% ± 3.90%** | **91.39%** | 7.67 |
| GCN-DAP | Wavelet | 78.98% ± 3.38% | 84.95% | 1.83 |
| GCN-ASAP | Inception V3 | 67.90% ± 11.09% | 82.80% | 9.67 |
| GCN-ASAP | Wavelet | 60.22% ± 5.63% | 72.04% | 2.30 |

Table 1: Performance evaluation of COVID-19 diagnosis and prognosis, where 'GCN-DAP' indicates the proposed GCN-based method integrated with the distance aware pooling. 'ASAP', 'DiffPool', and 'HGP-SL' refer to the state-of-the-art hierarchical pooling methods. Besides, we list the average time in seconds to complete one training epoch for each model using a single NVIDIA V100 GPU in the column 'Time (s/epoch)'.

**Diagnosis and prognosis performance.** According to the ROC Curve for diagnosis in Figure 4, our method is comparable to the senior radiologists with 15 to 25 years of clinical experience. In Table 1, we compared the performance of our model and GCN model with the state-of-the-art hierarchical pooling methods, including ASAP, DiffPool (Ying et al., 2018), and HGP-SL (Zhang et al., 2019). The maximum, average, and standard deviation of test accuracies on 20 random seeds are reported. We observed that the Inception V3 feature extraction method constantly outperforms

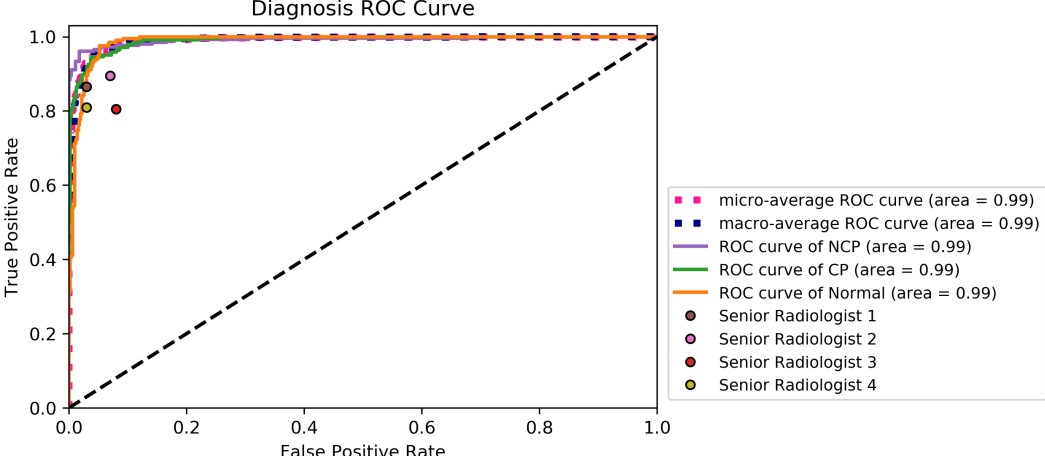

Figure 4: The ROC urve and AUC for diagnosis. 'NCP', 'CP', and 'Normal' indicate COVID-19 positive patients, common pneumonia patients, and healthy individuals respectively. The filled dots represent the performance on 'NCP' diagnosis of senior radiologists with 15 to 25 years of clinical experience (Zhang et al., 2020). It shows that our method is comparable to the senior radiologists.

the wavelet decomposition method under the same model configuration. The gradient explosion occurs in around 50% of the runs under the ASAP and DiffPool, resulting in optimization failures. ASAP and DiffPool are very unstable unfortunately, while this issue has not been witnessed during the training of our method and HGP-SL. Thus, the standard deviations of ASAP and DiffPool are much higher. Additionally, our method outperforms HGP-SL marginally but the training of our method is about 2 times faster.

The training curves using DAP versus the aforementioned hierarchical pooling methods over 20 runs, with varied random seeds and different train-validation-test split are presented in Figure 5. The figure shows that our model improves the training convergence, and DAP consistently outperforms ASAP and DiffPool across almost all runs.

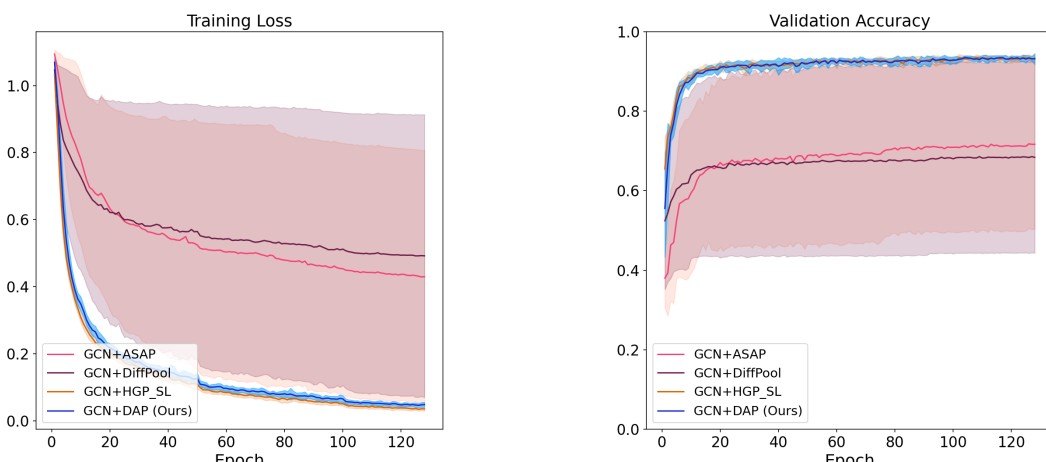

Figure 5: Training curves of GCN diagnosis model using DAP versus three hierarchical pooling methods over 20 runs, with varied random seeds and different train-validation-test split. The solid lines represent the mean training loss and validation accuracy, and the shade visualizes the interval of one standard deviation. It shows that DAP consistently outperforms ASAP and DiffPool across almost all runs, and converges much faster.

**Weakly supervised lesion localization results.** Besides graph classification, we also use our model to localize the most informative CT slices for each CT scan in the test dataset using the procedure in Section 3.3 with $k^s = 10$. Since the chosen CT slices are the most decisive ones for the diagnosis, they may contain lesions related to COVID-19 or common pneumonia. Then, we compare our

selected slices to the CT slices with lesions labeled by the dataset for each CT scan. Considering that the CT slices in the same CT scan are sequential, we choose the slices between the first and last slices localized by our model or labeled with lesions in the dataset. The localization of a COVID-19 patient is visualized in figure 6. The results are evaluated by Precision, Recall, and the Intersection over Union (IoU) for each patient, where IoU $= \frac{\#[\text{Localized Slices} \cap \text{Slices w/ Lesion}]}{\#[\text{Localized Slices} \cup \text{Slices w/ Lesion}]}$. Through hyperparameter searching, the best performance is achieved when the pooling proportion $k$ is 0.8. We reported the average and standard deviation of precision, recall and IoU among all patients in the test set on 20 random seeds in Table 2. The average IoU is 41.75% with standard deviation 2.59%.

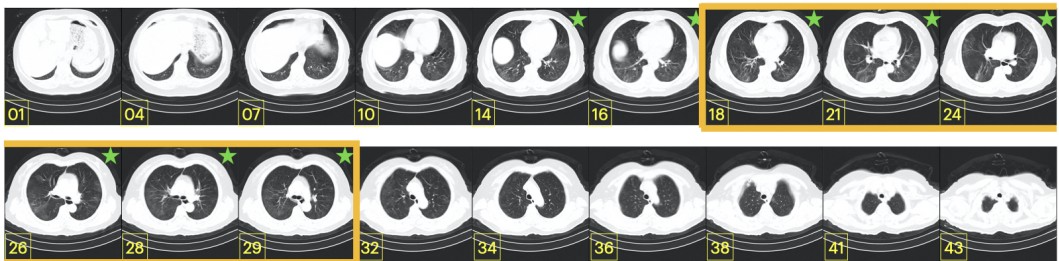

Figure 6: Visualisation of one-drop localization for a COVID-19 positive patient: The index at the bottom left corner of each slice refers to the slice index in the original CT scan. The slices with green stars at the top right corner are the slices containing lesions. The slices within the orange box are localized using the GCN diagnosis model. The recall, precision, and IoU of localization for this patient is 80%, 100%, and 80%, respectively.

| Average Precision $\pm$ SD | Average Recall $\pm$ SD | Average IoU $\pm$ SD |
|---|---|---|
| 57.39% $\pm$ 3.32% | 79.89% $\pm$ 3.94% | 41.75% $\pm$ 2.59% |

Table 2: Performance of one-drop localization. The localization results are compared with the CT slices containing lesions. We list the average and standard deviation of precision, recall, and IoU among patients in the test sets over 20 runs.

## 5    DISCUSSION

In this work, we introduce a framework that can provide accurate graph classification based on chest CT scans. Unlike previous diagnosis methods based on CT scans, our model can produce coarse localization highlighting the potential slices with lesions. To build trust in this framework and move towards clinical use, we should ensure that the model can explain the reason for their prediction instead of merely outputting the result. We argue that localization can help analyze prediction failure, and help researchers understand the effect of adversarial attacks in the medical imaging domain.

As one of the few lesion localization frameworks out there, we argue that the localized slices may help uncover how a given classification method arrives at the conclusion. Specifically, it might help reveal if a method is merely overfitting on the training data by examining the slices it attends to. This will help build more robust and explainable medical AI models. Conversely, it may also be used to identify the intrinsic dataset bias, notably the data acquisition bias (Biondetti et al., 2020), thereby guiding the dataset collection process and drive the medical imaging field forward.

## 6    CONCLUSION

This paper introduces an efficient and robust GCN-based diagnosis and prognosis system with a distance aware pooling method. It can cluster nodes and learn the patient-level representation hierarchically. Apart from accurate diagnosis, a weakly supervised lesion localization method is proposed to support prognosis, making the clinical decision more interpretable and reliable.

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
