# OpenReview forum: "Beyond COVID-19 Diagnosis: Prognosis with Hierarchical Graph Representation Learning"
_ICLR.cc/2021/Conference — Reject_

### Official Review · AnonReviewer4 · 2020-10-23
**Relevant to Covid-19 the current need, relevant to conference application track (GCN) - the paper is easy to read but core novelty is a bit low wrt ICLR's standards.**

**Rating:** 6
**Confidence:** 4

**Review:**

The paper is an application of GCN with good features on chest CT scan images for Covid-19 diagnosis and prognosis. First of all this is a relevant and appreciated effort when the world is fighting the pandemic. Hence some bonus points is directed towards that. As a whole, to the representation learning community, it adds limited research values apart from being an application of GCN which is aligned to the application track of ICLR. The paper claims that with less than 1% number of total parameters in the baseline 3D ResNet model, their method achieves 94.7% accuracy for diagnosis, which is marginally better than state of art - however whether the model was over-fitted is not clear. Prognosis information is an added claim, though automation part is not integrated.
It will be better if approach is compared with another concurrent work - "Covid-19 Classification by FGCNet with Deep Feature Fusion from Graph Convolutional Network and Convolutional Neural Network" - https://www.sciencedirect.com/science/article/pii/S1566253520303705.

I found too many repeating verbatim and technical descriptions - most can be delegated to references / citations or section numbers.

Sections 2.2 and 2.3 is not needed as such - if any special benefit wrt to current problem is given it will be helpful - more text from appendix can be added to paper than spending on explanations of easily searchable material.

Fig. 1 is too simple - go deep - can get influenced by similar works in state of art papers - how they present.

"Empirically, it is shown to be more robust for densely connected graphs" - this claim needs more substance.

"may be bootless" - use inefficient.

In Node clustering, how is 'k' determined?

As a general comment, too many to point individually - break the sentences around mathematical terminology into parts and also check the flow of grammar.

"Haar wavelet function with resolution 3" - more details on this.

I am not sure about the ablation studies and following of Train-Eval-Test based experiments (refer Andrew Ng ML Yearning).

Conclusion - write with quantitative numbers and a few sentences on author's learning / observations from overall experiments / method. And to conclude from my side, the paper is easy to read, needs some polishing and quantitative explained presentation; but overall is a relevant application paper.

---

> ### Author Response · Authors · 2020-11-16
> **Response to Reviewer 4 [Part 1]**
>
> Thanks a lot for your insightful comments. We have given our responses as the follows:
>
> 1. *Whether the model was over-fitted is not clear*
>
>    Following your advice, we further provide the training curve in Figure 4. Figure 4 represents the training curves of the GCN diagnosis model using DAP versus ASAP over 20 runs with varied random seeds and different train-validation-test split. The solid lines represent the mean training loss and validation accuracy, and the shade visualizes the interval of one standard deviation. It shows that DAP consistently outperforms ASAP across almost all runs, and converges much faster.
>
>
> 2. *Prognosis information is an added claim, though the automation part is not integrated.*
>
>   The prognosis is also an end-to-end model. The initialization of the prognosis model is based on the model trained for diagnosis.
>
>
> 3. *It will be better if approach is compared with another concurrent work - "Covid-19 Classification by FGCNet with Deep Feature Fusion from Graph Convolutional Network and Convolutional Neural Network" -* [https://www.sciencedirect.com/science/article/pii/S1566253520303705*](https://www.sciencedirect.com/science/article/pii/S1566253520303705)*.*
>
>    We note that this paper is published on October 9th, which is later than the deadline for ICLR. We will compare the above methods with our model. However, this may be finished in the discussion period. We will finish it as soon as possible and add it to the experiment section before publication.
>
>
> 4. *too many repeating verbatim and technical descriptions - most can be delegated to references / citations or section numbers.*
>
>    Thank you for your insightful advice. We have polished the wording accordingly.
>
>
> 5. *Sections 2.2 and 2.3 is not needed as such - if any special benefit wrt to current problem is given it will be helpful - more text from appendix can be added to paper than spending on explanations of easily searchable material.*
>
>    We have summarized the related work section accordingly.
>
>
> 6. *Fig. 1 is too simple - go deep - can get influenced by similar works in state of art papers - how they present.*
>
>    We have changed the Schema of our model structure accordingly and provide more description in the caption of Figure 1.
>
>
> 7. *"Empirically, it is shown to be more robust for densely connected graphs" - this claim needs more substance.*
>
>    We will provide more substance by comparing with other GCN methods. We think this issue will be solved with more method comparison, including DiffPool. We will update the results within the rebuttal period.
>
>
> 8. "*may be bootless" - use inefficient.*
>
>    We have changed it.
>
>
> 9. *In Node clustering, how is 'k' determined?*
>
>    For node cluster, the pooling ratio $k$ is determined empirically. The pooling ratio does not significantly influence graph classification and localization if ratio k is larger than 50%.
>
>
> 10. *As a general comment, too many to point individually - break the sentences around mathematical terminology into parts and also check the flow of grammar.*
>
>    Thanks for your advice. We have polished our writing.
>
>
> 11. *"Haar wavelet function with resolution 3" - more details on this.*
>
>    Haar wavelet is a sequence of rescaled functions, which can be used in image compression. Resolution 3 means that the image decomposition processes are performed three times. After 3 decomposition processes, the approximation coefficients, which contain most information of the image, are used for downstream calculations.

---

> > ### Author Response · Authors · 2020-11-16
> > **Response to Reviewer 4 [Part 2]**
> >
> >
> > 10. *I am not sure about the ablation studies and following of Train-Eval-Test based experiments (refer Andrew Ng ML Yearning).*
> >
> >     For dataset 19nCoVR, 60% of individuals' CT scans are randomly chosen as the training set, 25% as the test set, and the remaining 15% for the validation. To avoid information leakage, the dataset is split according to individuals instead of the CT scan. We repeat the data splitting aforementioned on 20 random seeds. For each random seed, the model is trained from scratch. The maximum, average, and standard deviation of test accuracies are reported on page 7 during the rebuttal.
> >
> >     The detailed results are attached below.
> >
> >     (a) Diagnosis Performance evaluation
> >
> >     -----------------------------------------------------------
> >
> >     Method			Feature Extractor		Average Accuracy $\pm$ SD			Best Accuracy
> >
> >     GCN-DAP 		Inception V3 				93.93% $\pm$ 0.41%   						94.80%
> >
> >     GCN-DAP 		Wavelet   					 83.65% $\pm$ 1.01%    						85.16%
> >
> >     GCN-ASAP   	 Inception V3    			75.20% $\pm$ 18.70%     					93.74%
> >
> >     GCN-ASAP		Wavelet   					 51.43% $\pm$ 11.90%    					  81.50%
> >
> >     -----------------------------------------------------------
> >
> >     (b) Prognosis Performance evaluation
> >
> >     Method			Feature Extractor		Average Accuracy $\pm$ SD			Best Accuracy
> >
> >     GCN-DAP 		 Inception V3				82.70% $\pm$ 3.90%						91.39%
> >
> >     GCN-DAP		  Wavelet						78.98% $\pm$ 3.38%						84.95%
> >
> >     GCN-ASAP		Inception V3				67.90% $\pm$ 11.09%					  82.80%
> >
> >     GCN-ASAP		Wavelet  					  60.22% $\pm$ 5.63%						72.04%
> >
> >     -----------------------------------------------------------
> >
> >
> >
> > 11. *write with quantitative numbers and a few sentences on author's learning / observations from overall experiments / methods.*
> >
> >     We have further added it and a discussion part accordingly.

---

> > > ### Comment · AnonReviewer4 · 2020-11-24
> > > **Response to Author(s) from Anon Reviewer 4**
> > >
> > > Most of my comments has been addressed in the revision. Hence I have upgraded my rating to honor the effort. However, I still  feel that the core novelty wrt ICLR standard is something that can't be overcome at this moment. Waiting for other reviewers to discuss this. This is a good work.

---

### Official Review · AnonReviewer3 · 2020-10-27
**Recommendation to Reject**

**Rating:** 4
**Confidence:** 4

**Review:**

Summary
The manuscript proposes  a distance aware pooling method to use in  graph convolutional neural for predicting whether a subject is infected with Covid-19 (diagnosis) and progression of the disease (prognosis). Experiments were conducted on CT images from three groups: Covid-19 group, common pneumonia group, and heathy group with about 900 samples in each group. The proposed model achieved 94.7% accuracy.

Pros:

The manuscript proposes  a distance aware pooling method which, with the help of features generated through inception v3, could achieve an error rate of a little bit over 5%


Cons:

In terms of the potential impact of the work, my concerns are as follows:
1.	Although the model trained on the dataset achieved a relatively low error rate (~ 5.3% ),  there is no evidence that this error rate is low enough for the model developed to be really useful, e,g. could actions taken based on a diagnosis method of this level of error rate potentially create a local epidemic?
2.	The model is not tested on an independent dataset that consists of other type of lung diseases other than common pneumonia. Is it possible that the model could actually categorize CT from a patient with another type of lung disease as Covid-19?  Thus, this 94.7% accuracy probably needs to be taken with a grain of salt when the model is to be used to test patients clinically.

In terms of the technical soundness of the work and writing of the manuscript, my concerns are as follows.

1.	Although the manuscript claims the distance aware pooling method as its main contribution, the description of the method is no more than one figure. It is unclear how this pooling function is computed exactly. Nor is it clearly stated how does it fit in overall model of the graph convolutional neural network as well. Even in the figure, some of the terms is used without giving clear definition (e.g. adjacency matrix centered at a specific node ). Further, There is no discussion of strengths and limitations of the proposed pooling method and the method it is trying to compare against ASAP, specifically in the context of CT imagining.
2.	Almost the same can be said about the section of “improved receptive field”.  What is the motivation of improving receptive field? Why is it relevant in the case of graph formed of CT imaging. (incidentally, it seems that how authors do not mention in the manuscript how the edges are added to the graph, but I might have missed this). The impact of the radius “h_d” on the model performance is  not discussed, either.
3.	How many time have the authors randomly split training and testing set? the variance from different runs are not reported. It is a standard practice to report both the mean of the performance from different runs as well as the variance.
4.	Since the problem that the manuscript is trying to solve is a graph classification and main contribution of the manuscript is a pooling method, the authors should consider comparing their pooling methods with different pooling methods and using different graph classification methods as baseline as well.
See
Zhang et al   Hierarchical Graph Pooling with Structure Learning      https://arxiv.org/abs/1911.05954
Li et al   Graph Matching Networks for Learning the Similarity of Graph Structured Objects      ICML 2019

Overall, the manuscript could be improving by stating more clearly the motivation of their pooling method in the context of CT imaging,  giving a more complete and rigorous definition of the pooling function,   comparing their method against different pooling methods  using different classification methods,  and having a more detailed discussion on the strengths and limitations of different pooling methods.

---

> ### Author Response · Authors · 2020-11-16
> **Response to reviewer 3 [Part 1]**
>
> We appreciate your kind and insightful comments.
>
> The following are responses to detailed comments.
> 1. *Although the model trained on the dataset achieved a relatively low error rate (~ 5.3% ), there is no evidence that this error rate is low enough for the model developed to be really useful, e,g. could actions taken based on a diagnosis method of this level of error rate potentially create a local epidemic?*
>
>    Although RC-PCR is the standard diagnostic method, it has several limitations, especially its low sensitivity. According to Fang et al., 2020, the sensitivity of chest CT is much greater than that of RT-PCR (98% vs 71%, respectively; P > 001). Our model’s sensitivity of COVID-19 diagnosis is 93.8%, which shows that our work is a great alternative to RT-PCR and can be clinically useful.
>
>    Fang, Y., Zhang, H., Xie, J., Lin, M., Ying, L., Pang, P., & Ji, W. (2020). Sensitivity of Chest CT for COVID-19: Comparison to RT-PCR. Radiology, 296(2). doi:10.1148/radiol.2020200432
>
>    We used the datasets and baseline 3D-ResNet published by Zhang et al. (2020) for our work. For diagnosis tasks, the baseline method is superior to junior radiologists and comparable to mid-senior radiologists. Furthermore, our proposed method shows even better results than Zhang et al.'s work for diagnosis and prognosis. Thus, we believe that our method could assist clinicians and alleviate the burdens on the healthcare systems. Our method outperforms the recent counterpart consistently and significantly. Our method achieves 94% average test accuracy with only 0.41% standard deviation, while the average accuracy of baseline GCN-ASAP is 75% with a 19% standard deviation. We further provided the maximum, average, standard standard deviation of test accuracies over 20 runs, with various random seeds and different train-validation-test splits.
>
>    The training curve using DAP versus ASAP over 20 runs are presented in Figure 4 in our paper. The figure shows that our model improves the training convergence. It shows that DAP consistently outperforms ASAP across almost all runs.
>
>    Below are the detailed experiment results over 20 runs, with various random seeds and different train-validation-test splits.
>
>    (a) Diagnosis Performance evaluation
>
>    -----------------------------------------------------------
> Method			Feature Extractor		Average Accuracy $\pm$ SD			Best Accuracy
>
>    GCN-DAP 		Inception V3 				**93.93% $\pm$ 0.41%**   						94.80%
>
>    GCN-DAP 		Wavelet   					 83.65% $\pm$ 1.01%    						85.16%
>
>    GCN-ASAP   	 Inception V3    			75.20% $\pm$ 18.70%     					93.74%
>
>    GCN-ASAP		Wavelet   					 51.43% $\pm$ 11.90%    					  81.50%
>
>    -----------------------------------------------------------
>
>    (b) Prognosis Performance evaluation
> Method			Feature Extractor		Average Accuracy $\pm$ SD			Best Accuracy
>
>    GCN-DAP 		 Inception V3				82.70% $\pm$ 3.90%						91.39%
>
>    GCN-DAP		  Wavelet						78.98% $\pm$ 3.38%						84.95%
>
>    GCN-ASAP		Inception V3				67.90% $\pm$ 11.09%					  82.80%
>
>    GCN-ASAP		Wavelet  					  60.22% $\pm$ 5.63%						72.04%
>
>    -----------------------------------------------------------
> 2. *The model is not tested on an independent dataset that consists of other types of lung diseases other than common pneumonia. Is it possible that the model could actually categorize CT from a patient with another type of lung disease as Covid-19? Thus, this 94.7% accuracy probably needs to be taken with a grain of salt when the model is to be used to test patients clinically.*
>
>    We used the datasets and baseline 3D-ResNet published by Zhang et al. (2020) for our work. For diagnosis tasks, the baseline method is superior to junior radiologists and comparable to mid-senior radiologists. Furthermore, our proposed method shows even better results than Zhang et al.'s work for diagnosis and prognosis. Thus, we believe that our method could assist clinicians and alleviate the burdens on the healthcare systems. Currently, we focus on the classification of healthy, common pneumonia, and Covid-19. The common pneumonia group consist of viral pneumonia, bacterial pneumonia, and mycoplasma pneumonia, all of which are the most common causes of pneumonia in China. We also agree that further collaborations for collecting more data of lung diseases other than pneumonia will further improve our result.
>
>
> For more responses, please see Response to reviewer 3 [Part 2]

---

> > ### Author Response · Authors · 2020-11-16
> > **Response to reviewer 3 [Part 2]**
> >
> > 4. *Almost the same can be said about the section of “improved receptive field”. What is the motivation of improving receptive field? Why is it relevant in the case of graph formed of CT imaging. (incidentally, it seems that how authors do not mention in the manuscript how the edges are added to the graph, but I might have missed this). The impact of the radius “h_d” on the model performance is not discussed, either.*
> >
> >    Many thanks for your questions on Improved Receptive Field. The motivation of creating improved receptive fields originates from the work of Ranjan et al. (2020). In their work, they defined $RF^{node}$ as the number of hops required to cover the neighborhood of a given node. However, since $RF^{node}$ can only be integers, for a densely connected weighted graph, $RF^{node}$ may be insufficient because even given a small value of h, for instance, h =1, clusters formed within h = 1 may include most of nodes in a graph. Therefore, we introduce the improved receptive field, which accepts real values and utilise the image similarity information as well. We defined the edge as the cosine similarity of two CT slices, which measure the orientation of the node embedding instead of the magnitude.
> >
> >
> >
> > 5. *How many times have the authors randomly split training and testing sets? The variance from different runs are not reported. It is a standard practice to report both the mean of the performance from different runs as well as the variance.*
> >
> >    Thanks for your advice!
> >
> >    We randomly split the set into train/validation/test based on patient ID to ensure that that all CT volumes of one patient are in the same subset. This is to prevent information leakage. And during the training, the validation set is used for hyper-parameter tuning and model selection. Besides, variance and mean will also be reported for all trials accordingly. For a more accurate and fair comparison of methods, we split the datasets randomly twenty times with different random seeds. Then, we report the maximum, average, and standard deviations of test accuracies over 20 runs, with varied random seeds and different train-validation-test split during the rebuttal. The table is also presented in the response to Question 1.
> >
> >
> >
> > 6. *Since the problem that the manuscript is trying to solve is a graph classification and main contribution of the manuscript is a pooling method, the authors should consider comparing their pooling methods with different pooling methods and using different graph classification methods as baseline as well. See Zhang et al Hierarchical Graph Pooling with Structure Learning* [*https://arxiv.org/abs/1911.05954*](https://arxiv.org/abs/1911.05954) *Li et al Graph Matching Networks for Learning the Similarity of Graph Structured Objects ICML 2019*
> >
> >    Yes, we agree with you. We are working on the comparison with other recent graph classification methods, and will report the results as soon as possible.

---

> ### Author Response · Authors · 2020-11-25
> **Response to reviewer 3 - Comparison to other Hierarchical Pooling Methods**
>
> Thanks very much for your insightful comments.
>
> Regarding your concern #6, we further compare our pooling method with two more state-of-the-art pooling methods, DiffPool and HGP-SL. The maximum, average, and standard deviation of test accuracies on 20 random seeds are reported. Our method constantly outperforms ASAP and DiffPool over 20 runs. The gradient explosion occurs in around 50% of the runs under the ASAP and DiffPool, resulting in optimization failures. Unfortunately, they are very unstable, while this issue has not been witnessed during the training of our method and HGP-SL. Thus, the standard deviations of ASAP and DiffPool are much higher. Additionally, our method outperforms HGP-SL marginally, but the training of our method is about 2 times faster.
>
> Below is the detailed results. We also list the average time in seconds to complete one training epoch for each model using a single NVIDIA V100 GPU  in the last column, 'Time(s/epoch)'.
>
>  Diagnosis Performance evaluation
>
> | Method       | Feature Extractor | Average Accuracy $\pm$ SD | Best Accuracy | Time (s/epoch) |
> | ------------ | ----------------- | ------------------------- | ------------- | -------------- |
> | GCN-DAP      | Inception V3      | 93.93% $\pm$ 0.41%        | 94.80%        | 22.70          |
> | GCN-DAP      | Wavelet           | 83.65% ± 1.01%            | 85.16%        | 20.90          |
> | GCN-ASAP     | Inception V3      | 75.20% ± 18.70%           | 93.74%        | 30.00          |
> | GCN-ASAP     | Wavelet           | 51.43% ± 11.90%           | 81.50%        | 27.25          |
> | GCN-DiffPool | Inception V3      | 71.22% $\pm$ 23.73%       | 94.31%        | 18.35          |
> | GCN-DiffPool | Inception V3      | 93.89% $\pm$ 0.39%        | 94.22%       | 45.60          |

---

### Official Review · AnonReviewer1 · 2020-10-28
**Interesting application but lacking novelty**

**Rating:** 6
**Confidence:** 4

**Review:**

Short summary
---------------------
The authors propose a graph convolutional network (GCN) approach to perform the diagnosis and prognosis of COVID-19 from chest CT scans. They propose a novel pooling that takes into account the edges weights compared to recent methods. They validate their method on one dataset and compare the results to one recent baseline.

Strengths
--------------
The paper is very well written and easy to follow. The figures and tables are clear and pleasing. The work is sound.

Weaknesses
-------------------
While the motivation is clearly depicted, I lacked the clinical relevance or justification to tackle this problem. In addition, case analyses or the usefulness of the proposed prognosis tool are not discussed, and it is difficult to estimate the complexity of the task without this information (e.g. how clinicians evaluate severity, is it obvious from the pulmonary damage?).
On the other hand, the technical contribution is underwhelming: the proposed method is only marginally better than its recent counterpart, no confidence intervals are provided and the effect of the preprocessing is much larger. It is also unclear what the motivation was for some choices, how these relate to the literature or how they were practically implemented (see detailed comments).

Novelty
-------------
I did not find the technical contribution of significance, especially given the results comparing DAP to ASAP. While the problem of COVID-19 diagnosis and prognosis is timely, it lacked clinical insights.

Clarity
-----------
Very clear, very well written. A couple of details are missing (see detailed comments), but overall this is an enjoyable read. I wished there was a discussion of the technique, the results and the limitations.

Rigor
--------
The methods seemed sound. I wished confidence intervals were provided.

Detailed comments
---------------------------
- node clustering: can parts of the graph be “dropped” during this step?
- If the adjacency is already defined as the cosine distance between nodes, isn’t message passing already doing/reinforcing some kind of clustering? How does the clustering ranking (based on A) relate to comparing nodes in “input” space compared to the layer at stake? Given the proposed next layer connectivity, isn’t the next layer A an exponent version of the first layer A for the terms in top k?
- How to define h_d and k? What is their effect on model performance?
- Is there a justification behind the aggregation? Is there a reference for this? Doesn’t that strategy assume that each feature in 1...D’ across nodes N’ represent the same dimension?
- How would this compare to using a “master node” (Gilmer et al., 2017), where all nodes could write and read from a separate node, to allow for long-distance communication?
- From my understanding, one-drop localization is similar to occluding a cluster at a time. Is there a reference for this method?
- How is k_s chosen? Why is a fixed number chosen compared to e.g. a “% of prediction variance explained” type of approach?
- Any clinical input in this work?
- Is the number of slices acquired or selected through sampling mentioned in the text? This would help in illustrating the size of each graph.
- What are the reasons behind the choice of the 2 preprocessing techniques? They seem arbitrary without referencing.
- On such a low number of patients, confidence intervals are strongly recommended. I am not confident about the significance of DAP compared to ASAP, especially given the large performance gap between feature extractors.
- Comparison with CNNs would be interesting. This is especially important given the impact of the feature representation. I am wondering whether GNNs are not “too sensitive” to the choice of initial node embedding (which also defines A) and whether this should be investigated in more depth, or maybe CNNs display similar performance without needing this additional step.
- Won’t IoU on slices heavily depend on the choice of k_s?
- Selecting slices “included” in the set of slices highlighted by one-drop seems like an arbitrary choice to me. The fact that slices are sequential is specific to this type of data, and this choice seems like an “a posteriori” choice that better aligns with the ground truth.
- IoU is not reported across all patients, but rather on cherry-picked examples  in Table 2.
- There is no discussion

Minor
-------
- One-drop: I am confused on the wording, where the authors mention the score on the “target class”. Do they refer to the predicted class of the model?
- The prognosis task is undefined

---

> ### Author Response · Authors · 2020-11-16
> **Response to reviewer 1 [Part 1]**
>
> We appreciate your kind and insightful comments.
>
> We agree that the clinical insights involved in this work play a vital role in deciding our method's usefulness and impact.  In fact, we are working with a medical doctor to expand our method to incorporate survival data and progression information of CT imaging over time. We used the datasets and baseline 3D-ResNet published by Zhang et al. (2020) for our work. For diagnosis tasks, the baseline method is superior to junior radiologists and comparable to mid-senior radiologists. Furthermore, our proposed method shows even better results than Zhang et al.'s work for diagnosis and prognosis. Thus, we believe that our method could assist clinicians and alleviate the burdens on the healthcare systems. Our collaborators with clinical background and some physicians stated that this severity assessment of a patient’s condition plays a vital role in the patient management, and predicting the need for ICU or mechanical ventilation in advance can also help to plan the management or prepare the patient better, especially when the medical resources are limited. Our work indeed receives positive feedbacks and support from clinicians.
>
> During the rebuttal, we further report the average and standard deviation of test accuracy over 20 runs, with varied random seeds and different train-validation-test split, for diagnosis, prognosis apart from the best accuracy for each model. Also, the average IoU of the weakly supervised localization is provided in the manuscript.
>
> Below are the detailed results,
>
> (a) Diagnosis Performance evaluation
>
>
> Method							Feature Extractor				Average Accuracy $\pm$ SD					Best Accuracy
>
> GCN-DAP 						Inception V3 						93.93% $\pm$ 0.41%   								94.80%
>
> GCN-DAP 						Wavelet   					 		83.65% $\pm$ 1.01%    								85.16%
>
> GCN-ASAP   	 				Inception V3    					75.20% $\pm$ 18.70%     							93.74%
>
> GCN-ASAP						Wavelet   							 51.43% $\pm$ 11.90%    					  		81.50%
>
>
>
> (b) Prognosis Performance evaluation
>
> Method					Feature Extractor				Average Accuracy $\pm$ SD					Best Accuracy
>
> GCN-DAP 				 Inception V3						82.70% $\pm$ 3.90%								91.39%
>
> GCN-DAP		  		Wavelet								78.98% $\pm$ 3.38%								84.95%
>
> GCN-ASAP				Inception V3						67.90% $\pm$ 11.09%					  		82.80%
>
> GCN-ASAP				Wavelet  					  		60.22% $\pm$ 5.63%								72.04%
>
>
> According to the results above, we can see that our method indeed consistently outperforms ASAP with much smaller standard deviations. Meanwhile, the effect of proposed method is larger than the preprocessing. Compared with the large standard deviation of GCN+ASAP, our method is much more precise and accurate. The training curves of our methods and baseline model is also presented in the manuscript, which validate the robustness of our method. It shows that our method converges much faster.
>
> (c) For the weakly-supervised localization, we further report the average IoU. We regard the localization method as a visual explanation for prognosis and diagnosis results. Thus, they may be highly correlated with the CT slices containing lesions. Thus, we further report the average IoU, which is 41.75% with a standard deviation of 2.59%.
>
> For technical contributions, we first proposed a distance pooling method, which is more robust for densely connected weighted graphs. Besides, we proposed the one-drop localization method, which only utilized the label for diagnosis. Thus, we are able to integrate the localization method to the diagnosis and prognosis model in an end-to-end fashion. In this paper, we regard the localization method as a visual explanation for diagnosis and prognosis results.
>
> Zhang, K., Liu, X., Shen, J., Li, Z., Sang, Y., Wu, X., ... & Ye, L. (2020). Clinically applicable AI system for accurate diagnosis, quantitative measurements, and prognosis of covid-19 pneumonia using computed tomography. Cell.
>
> For the response to the detailed comments, please see Response to reviewer 1 [Part 2, Part 3, Part 4, and Part 5]

---

> > ### Author Response · Authors · 2020-11-16
> > **Response to reviewer 1 [Part 2]**
> >
> > Below is the response to your detailed comments:
> >
> > 1. *Case analyses or the usefulness of the proposed prognosis tool are not discussed; need more perspectives from clinicians to judge the usefulness. Lack the clinical relevance or justification to tackle this problem.*
> >
> >    Our collaborators with clinical background and some physicians stated that this severity assessment of a patient’s condition plays a vital role in patient management, and predicting the need for ICU or mechanical ventilation in advance can also help to plan the management or prepare the patients for better management of the disease, especially when the medical resources are limited. When acquiring the data from clinicians and medical professionals, we noted the pressing needs of localizing the lesions. Therefore, we designed the solution with the goal of both accurate classification and interpretability.
> >
> > 2. *The proposed method is only marginally better than its recent counterpart.*
> >
> >    Our work involves both diagnosis and prognosis in an integrated manner. The methods in the literature appear to focus on one aspect only, either diagnosis or prognosis. Our method outperforms the state-of-the-art diagnosis, 3D-ResNet, in the literature. But, we only use about 1% parameter. Instead of only reporting the classification result, our model produces coarse localization highlighting the potential slices with lesions, which could interpret the decision. To build trust in this framework and move towards clinical use, we should ensure that the model can explain the reason for their prediction instead of merely outputting the result. We argue that localization can help analyze prediction failure, and help researchers understand the effect of adversarial attacks in the medical imaging domain.
> >
> > 3. *No confidence intervals are provided.*
> >
> >    To give a more complete and fair comparison, we follow your suggestions and further report the mean and standard deviation of test accuracy on the 20 random seeds.
> > Our method achieves 94% average test accuracy with only 0.41% standard deviation, while the average accuracy of baseline GCN-ASAP is 75% with a 19% standard deviation. Therefore, we argue that our method outperforms GCN with ASAP consistently and significantly with the same preprocessing technique.
> >
> > 4. *The effect of the preprocessing is much larger.*
> >
> >    As shown in the table above, the effect of using our model is larger than that of the preprocessing technique.
> >
> > 5. *I did not find the technical contribution of significance, especially given the results comparing DAP to ASAP*
> >
> >    As mentioned above, our method outperforms the recent counterpart consistently and significantly with the same preprocessing technique. Our method achieves 94% average test accuracy with only 0.41% standard deviation, while the average accuracy of baseline GCN-ASAP is 75% with a 19% standard deviation. We added the results in the paper.
> >    The training curves using DAP versus ASAP over 20 runs, with various random seeds and different train-validation-test splits are presented in Figure 4 in our paper. The figure shows that our model improves training convergence. It shows that DAP consistently outperforms ASAP across almost all runs.
> >
> > 6. *It is also unclear what the motivation was for some choices, how these relate to the literature, or how they were practically implemented*
> >
> >    Thanks for your advice. We added more implementation details in the manuscripts.
> >
> > For other responses, please see the following parts

---

> > > ### Author Response · Authors · 2020-11-16
> > > **Response to reviewer 1 [Part 3]**
> > >
> > > 7. *Node clustering: can parts of the graph be “dropped” during this step?*
> > >
> > >    Yes. The node dropout is adopted during training to avoid overfitting.
> > >
> > > 8. *If the adjacency is already defined as the cosine distance between nodes, isn’t message passing already doing/reinforcing some kind of clustering? How does the clustering ranking (based on A) relate to comparing nodes in “input” space compared to the layer at stake? Given the proposed next layer connectivity, isn’t the next layer A an exponent version of the first layer A for the terms in top k?*
> > >
> > >    Thank you for your insightful comments!
> > >    As CT scans may embody feature representations at slice-level,  lesion-level, and patient-level, the atomic modules of our model consist of a GCN layer and a pooling layer, as shown in Figure 1, to better model the hierarchical feature representations. The message passing of GCN is to propagate the information and learn node embeddings. However, it may lack the capability to abstract the topology of a graph in a hierarchical manner. Hence, we adopted pooling layers to cluster nodes and compute the hierarchical representation with reduced nodes in the graph. The GCN layer that follows the pooling layer can then learn the embedding of the clustered graph with coarser representation, which is similar to CNN and its pooling operations. In this fashion, the patient-level representation is learned and passed to the MLP layer for graph classification.  Ranjan et al. (2020) prove hierarchical feature aggregation is important for graph classification. We find this a reasonable extension to the CT classification setting as CT scans also embody multi-level features.
> > >
> > >    The dimension of the adjacency matrix for the input space is $N \times N$, where N is the number of nodes. After one GCN and pooling layer, the number of nodes is reduced to kN. Thus, the adjacency matrix of this layer has a size of $kN \times kN$, which is corresponding to a higher level graph structure. Similarly, the final adjacency is $k^l N \times k^l N$, where l is the number of layers. Through several GCN-pooling modules, the graph structure and embedding are learned interactively and hierarchically.
> > >
> > >    Ying, Z., You, J., Morris, C., Ren, X., Hamilton, W., & Leskovec, J. (2018). Hierarchical graph representation learning with differentiable pooling. In Advances in neural information processing systems (pp. 4800-4810).
> > >    Ranjan, E., Sanyal, S., & Talukdar, P. P. (2020). ASAP: Adaptive Structure Aware Pooling for Learning Hierarchical Graph Representations. In AAAI (pp. 5470-5477).
> > >
> > > 9. *How to define h_d and k? What is their effect on model performance?*
> > >
> > >    Thanks for your questions. $h^d$ is the notation of improved receptive field $RF^d$, which is a positive real number and can be understood as the radius. The value k, as mentioned in the Node Clustering part, is a value to help us to choose the top k percent clusters based on cluster scores. The effect of $h^d$ is that since $h^d$ is a positive real value, the image similarity information can be used to help node clustering, while in the work of Ranjan et al. (2020), they set h as an integer value, which performed badly in densely connected weighted graphs. The effect of k is that the k proportion could affect the depth of GCN models. Besides, the choice of k also affects the performance of the one-drop localization method.
> > >
> > > 10. *Is there a justification behind the aggregation? Is there a reference for this? Doesn’t that strategy assume that each feature in 1...D’ across nodes N’ represent the same dimension?*
> > >
> > >    Yes. The node feature has the same dimension. We have not seen a reference for this, while it is intuitive to conduct the aggregation and the result supports it.
> > >
> > > 11. *How would this compare to using a “master node” (Gilmer et al., 2017), where all nodes could write and read from a separate node, to allow for long-distance communication?*
> > >
> > >    Thanks for your advice. we followed your advice, and are trying to adopt 'master node'. We hope this could be added to the comparison in our paper during the rebuttal period.
> > >
> > > For other responses, please see the following parts.

---

> > > > ### Author Response · Authors · 2020-11-16
> > > > **Response to reviewer 1 [Part 4]**
> > > >
> > > > 12. *From my understanding, one-drop localization is similar to occluding a cluster at a time. Is there a reference for this method?*
> > > >    Actually, we have not seen a similar visual explanation in graph neural networks. The idea of one-drop localization is actually inspired by Class Activation Mapping, the techniques for visualizing which part CNN is looking at to make a decision (Zhou et.al, 2016). But, for the methodology, we are inspired by classic stepwise regression (Efroymson, 1960;  Krzysztof et al., 1998), where the variable selection is integrated by an automatic procedure. One of the ideas is backward elimination. Thus, we occlude a cluster at a time, which is equivalent to delete one candidate variable. Then, we consider one cluster as the most statistically significant cluster if the lowest score of the target class is achieved due to the deletion of that cluster.
> > > >
> > > >    Efroymson, M. A. (1960). Multiple regression analysis. Mathematical Methods for Digital Computers. Ralston A. and Wilf, H. S., (eds.), Wiley, New York.
> > > >
> > > > Krzysztof J Cios, Witold Pedrycz, and Roman W Swiniarski. Data mining and knowledge discovery.InData mining methods for knowledge discovery, pp. 1–26. Springer, 1998
> > > >
> > > >    Zhou, B., Khosla, A., Lapedriza, A., Oliva, A., & Torralba, A. (2016). Learning deep features for discriminative localization. In Proceedings of the IEEE conference on computer vision and pattern recognition (pp. 2921-2929).
> > > >
> > > > 13. *How is k_s chosen? Why is a fixed number chosen compared to e.g. a “% of prediction variance explained” type of approach?*
> > > >
> > > >    We explain in the paper that $k^s$ here is the top $k^s$ nearest slices to the chosen centers.
> > > >
> > > > 14. *Any clinical input in this work?*
> > > >
> > > >    In addition to the comments in Q1, our collaborator from the clinical background also gives us professional suggestions.
> > > >
> > > > 15. *Is the number of slices acquired or selected through sampling mentioned in the text? This would help in illustrating the size of each graph.*
> > > >
> > > >    Each graph has 48 nodes. Here, one node means one slice from a CT scan. The medium of the number of slices in CT scans in the dataset is around 48. Thus, we initially sampled 48 for each CT scan.  We also compared the performance of using 24, 48, and 64 slices per CT scan. Using more than 48 slices for each graph cannot further improve the model performance. Thus, we specify 48 nodes for each graph for this paper.
> > > >
> > > > 16. *What are the reasons behind the choice of the 2 preprocessing techniques? They seem arbitrary without referencing.*
> > > >    Thanks for you
> > > > r question. Using the same model, we compare the effect of two preprocessing techniques. Feature extraction with InceptionV3 is a common strategy.
> > > > Meanwhile, wavelet decomposition extraction, with elegant mathematical expressions, is to provide an alternative explainable method. However, it is expected that wavelet decomposition extraction compromises the test accuracy since it extracts information from each image individually.
> > > >
> > > > 17. *On such a low number of patients, confidence intervals are strongly recommended. I am not confident about the significance of DAP compared to ASAP, especially given the large performance gap between feature extractors.*
> > > >
> > > >    Thanks for your advice. We provided the mean and standard deviation of test accuracy on 20 runs with different train-validation-test splits, as mentioned above. Actually, the gradient explosion occurs in around 50% of the runs under the ASAP model, resulting in optimization failures. ASAP is very unstable unfortunately, while this issue has not been witnessed during the training of our method. Thus, the standard deviation of test accuracy for ASAP is much higher, around 18%.
> > > >    The mean accuracy of ASAP is 75.20%, while DAP achieves 93.93% with a standard deviation of 0.41%. Thus, we argue that our method significantly and consistently outperforms the baseline. Also, in terms of convergence speed, our model performs much better.
> > > >
> > > > 18. *Comparison with CNNs would be interesting. This is especially important given the impact of the feature representation. I am wondering whether GNNs are not “too sensitive” to the choice of initial node embedding (which also defines A) and whether this should be investigated in more depth, or maybe CNNs display similar performance without needing this additional step.*
> > > >
> > > >    Our baseline CNN model is from Zhang et.al. (2020), which is 3D-ResNet, and our method outperforms it and requires around 1% number of parameters of the baseline. Our GCN model opens a new door to densely connected graphs, which, to our best knowledge, is the first application of GCN to CT images.
> > > >
> > > > For other responses, please see the following parts.

---

> > > > > ### Author Response · Authors · 2020-11-16
> > > > > **Response to reviewer 1 [Part 5]**
> > > > >
> > > > > 19. *Won’t IoU on slices heavily depend on the choice of k_s?*
> > > > >
> > > > >    We agree that the choice of $k^s$ will affect the average IoU. Actually, in real clinical cases, different radiologists may choose different CT slices of the same patient to do the diagnosis. We believe that when using our model,  the radiologists can adjust the value of $k^s$ based on the patient’s condition and their personal preference to assist the COVID-19 diagnosis.  We think the clinical significance of our model is to provide diagnosis suggestions with chosen slices as an important insight to the clinicians.
> > > > >
> > > > > 20. *IoU is not reported across all patients, but rather on cherry-picked examples in Table 2.*
> > > > >
> > > > >    Thanks for your advice. We consider the localization as a visual explanation for the decision made by the model. We reported the Average IoU across all patients by comparing the localization results with the slices containing lesions. The average IoU of the test set patients is 41.75% with a standard deviation of 2.59%.
> > > > >
> > > > > 21. *There is no discussion part.*
> > > > >
> > > > >    Thanks for your advice. Due to space limitations, we did not, while now we add a discussion part accordingly.
> > > > >
> > > > > 22. *MINOR 1: One-drop: I am confused on the wording, where the authors mention the score on the “target class”. Do they refer to the predicted class of the model?*
> > > > >
> > > > >  Yes. The score mentioned is the prediction score for the full model. Sorry for the confusion.
> > > > >
> > > > > 23. *MINOR 2: The prognosis task is undefined.*
> > > > >
> > > > > The definition of the prognosis task is defined in the problem statement and the clinical definition of prognosis is introduced in section 4.3 Dataset.
> > > > >    We regard the prognosis as a graph classification problem. For prognosis, the class indicates whether a COVID-19 positive patient develops into severe/critical illness status. Thus, there are two classes, mild and severe/critical illness status. As introduced in section 4.1 Dataset, the second class, severe/critical illness status refers to the admission to ICU, mechanical ventilation,  or death.

---

### Author Response · Authors · 2020-11-16
**Revised manuscript is uploaded**

We appreciate the reviewers' valuable comments and insightful feedback. Following reviewers' advice, we updated the manuscript. Below are the major revisions,

1. We adjusted our figures. For Figure1, we added more details and descriptions about the model configuration. For instance, we emphasize the relationship between the topological changes of graphs. For Figure 2, a numerical example is further provided to help explain our methodology. The processes mentioned in Figure 2 correspond to those mentioned the section 3.3.2 Distance Aware Pooling Method.
2. We added more details on model configuration and implementation in the section 4.3.
3. Apart from the best test accuracy, we further reported the average and standard deviation of test accuracies over 20 runs, with varied random seeds and different train-validation-test split. Also, the training curves are visualized in Figure 4. The average precision, recall, and IoU for weakly supervised localization is provided in the section 4.4
4. Apart from ASAP, we further compare our pooling method with two state-of-the-art hierarchical pooling methods, DiffPool and HGP-SL for the diagnosis task in section 4.4.
5. We provided the ROC Curve for diagnosis in Figure 4 to compare the performance of our model and senior radiologists with 15-25 years of clinical experience.
6. We added the discussion section accordingly to discuss the interpretability of our method in the field of medical imaging.

---

### Decision · Program_Chairs · 2021-01-07
**Final Decision**

**Decision:**

Reject

**Comment:**

The paper presents a GCN-based solution with a distance aware pooling method for diagnosis and prognosis of COVID-19 based on CT-scan. It aims to address an important and timely problem. The proposed solution is reasonable.

The paper receives mixed ratings, and therefore we had extensive discussions. It is agreed by all of us that

(1) the novel contribution of the proposed method is relatively low compared with standard ICLR papers;

(2) the evaluation is interesting, but could be improved with state-of-art baselines on CT-scan (not limited to GCN-based method);

(3) the authors have improved the writing of the paper significantly, which convinces two reviewers to elevate their scores.

The paper addresses a timely topic, but there is still room for improvement in methodology and evaluation. We hope that the reviews can help the authors prepare a strong publication in the future.